# Chemotaxis of *Clostridium* Strains Isolated from Pit Mud and Its Application in Baijiu Fermentation

**DOI:** 10.3390/foods11223639

**Published:** 2022-11-14

**Authors:** Langtao Wu, Jingya Fan, Jian Chen, Fang Fang

**Affiliations:** 1Science Center for Future Foods, Jiangnan University, Wuxi 214122, China; 2Key Laboratory of Industrial Biotechnology, Ministry of Education, School of Biotechnology, Jiangnan University, Wuxi 214122, China; 3Engineering Research Center of Ministry of Education on Food Synthetic Biotechnology, Jiangnan University, Wuxi 214122, China; 4Jiangsu Province Engineering Research Center of Food Synthetic Biotechnology, Jiangnan University, Wuxi 214122, China

**Keywords:** *Clostridium*, chemotaxis, motility, lactic acid, strong-flavor baijiu, pit mud

## Abstract

*Clostridium* is the key bacteria that inhabits the pit mud in a fermentation cell, for the production of strong-flavor Baijiu. Its activities in the process of Baijiu fermentation is closely related to the niches of pit mud and cells. After multiple rounds of underground fermentation, *Clostridium* has been domesticated and adapted to the environment. The mechanisms of *clostridia* succession in the pit mud and how they metabolize nutrients present in grains are not clear. In this study, 15 *Clostridium* species including three firstly reported ones (*Clostridium tertium, Clostridium pabulibutyricum* and *Clostridium intestinale*) in strong-flavor Baijiu pit mud, were isolated from the pit mud. Eighty one percent of these *Clostridium* strains are motile, and most of them show chemotaxis to organic acids, glutathione, saccharides and lactic acid bacteria. In a simulated Baijiu fermentation system, *Clostridium* migrated from pit mud to fermented grains with the addition of chemokine lactic acid, resulting in the production of acetic acid and butyric acid. The results help to understand the succession mechanism of *Clostridium* in pit mud, and provide a reference for regulation of lactic acid level in fermented grains during Baijiu fermentation.

## 1. Introduction

Strong-flavor Baijiu is a popular Baijiu in China. Its aroma and quality are greatly influenced by the physiochemical properties and the microbial composition of pit mud [1]. Microorganisms in the pit mud also have important influences on the formation of volatiles in Baijiu. *Clostridium* is a functional bacterium in pit mud that synthesizes butyric acid, caproic acid and other key volatiles from various substances such as sugars, fatty acids, alcohols and sulfates [2,3,4]. The composition, population and metabolic properties of *Clostridium* in pit mud are diverse, due to the heterogeneity and complexity of pit mud. There are about 39 species of *Clostridium* that have been isolated from strong-flavor Baijiu pit mud [4,5]. Characterization of the motility and chemotaxis of these *Clostridium* strains from the pit mud helps to understand their behavior in the process of fermentation and disclose their succession mechanism in the pit mud during the long-term fermentation.

During strong-flavor Baijiu fermentation, the microbial community of pit mud experiences a long-term (10–30 years) succession process (aging process) before it matures [6,7]. The motility and inhabitation of specific clostridia in pit mud might be triggered by chemotaxis to their chemokines including both nutrients and microorganisms. The physicochemical properties (such as ammonium, available phosphorus, available potassium, moisture and pH) of the pit mud are dynamic during the long-term brewing process. Changes in physicochemical properties of the pit mud play roles in promoting the enrichment of functional bacteria in pit mud [8,9,10]. Thus, the relative abundance of *Clostridium* in the mature pit mud is usually higher than the relative fresh ones. It was found that moisture affected both the pH of pit mud and the growth of microorganisms [11]. Appropriate content of ammonium nitrogen is important in maintaining the quality of pit mud and improving the quality of Baijiu, as ammonium is necessary for microbial growth and synthesis of various proteins and enzymes [12]. The presence of available phosphorus and potassium in pit mud is also beneficial to microbial growth [13]. Particularly, moisture, available phosphorus/potassium, pH, ammonium and humus were found to be positively correlated with the composition of *Clostridium* in pit mud [14,15]. In addition, lactic acid and acetic acid were found to be the driving factors for the evolution of pit mud microbiota, and can efficiently enrich *Clostridium* [16]. Thus, differences in physicochemical properties of pit mud shift the community of *Clostridium* in it to some extent. A microbial tracking experiment revealed that pit mud was a continuous source of anaerobes. Anaerobes including *Clostridium*, *Petrimonas*, *Sedimentibacter* and *Syntrophomonas* can continuously migrate from the pit mud to fermented grains during fermentation [17,18]. Microorganisms in the pit mud not only increase the microbial diversity of fermented grains, but also improve Baijiu aroma through the production of organic acids and linear alcohols [19]. Apart from the passive migration driven by liquid (yellow water) produced during Baijiu fermentation, motile microorganisms in the pit mud can migrate to fermented grains. It was found that flagellar assembly and chemotaxis were more significantly enriched in pit mud-associated *Clostridium* than those non-pit mud-associated ones [4]. Moreover, most *Clostridium* have flagella and are motile. Thus, it increases the possibility of *Clostridium* migration to fermented grains for participating fermentation through chemotaxis. However, the migration mechanism of motile bacteria in the fermentation cells that relates to chemotaxis has not been clarified. Which chemokine triggers the motion of *Clostridium* and how they influence bacterial metabolism and Baijiu fermentation also needs to be disclosed.

This work aims to isolate motile *Clostridium* strains from strong-flavor Baijiu pit mud, and characterization of their chemotaxis to various chemokines. The results will help to explain the migration behavior of *Clostridium* and its role in fermentation of strong-flavor Baijiu. It also provides fundamentals for the control or regulation of strong-flavor Baijiu fermentation.

## 2. Materials and Methods

### 2.1. Collection of Samples

Pit mud samples used in this study were collected from a distiller in Jiangsu province, China. A five-point sampling method was used to collect pit mud from the bottom and four walls of the cellar during succession (10 years) and maturity periods (30 years) [7,20], the mixed portions were then transferred into sterile bags and stored at −80 °C.

### 2.2. Isolation and Cultivation of Clostridium

The pit mud sample was mixed with sterile 0.85% (*w*/*v*) NaCl before inoculating into reinforced clostridial medium (RCM). Then the solution was diluted and spread on RCM agar for anaerobic cultivation at 37 °C for 48 h. Single colonies were picked and purified by streaking on RCM agar for three times. Individual strain was inoculated into RCM broth and cultivated anaerobically before storing them at −80 °C.

### 2.3. Identification of Bacteria

The bacterial genomic DNA was extracted using a bacterial genomic DNA isolation kit (Sangon Biotech, Shanghai, China). The 16S rDNA was amplified using the primers 27F (5′-AGAGTTTGATCCTGGCTCAG-3′) and 1492R (5′-GGTTACCTTGTTACGACTT-3′) [21]. The PCR products were purified and sent to Talen-bio (Wuxi, China) for sequencing. Then the sequences were submitted to NCBI for online BLAST comparison to identify strains.

### 2.4. Analysis of Clostridium Motility

The culture of *Clostridium* (5 µL, exponential growth phase) was spotted on a semi-solid (0.4% agar, *w*/*v*) RCM agar, and incubated anaerobically at 37 °C for 48 h. Motility of *Clostridium* was indicated by the diffusion of colonies. The movement of *Clostridium* was also verified by using concave slides and optical microscope.

### 2.5. Analysis of Chemotaxis of Clostridium

The culture of motile *Clostridium* (1 mL, exponential growth phase) was centrifuged at 6000 r·min^−1^ for 5 min, cell pellet was washed twice with 0.85% (*w*/*v*) NaCl, and resuspended at 1 × 10^8^ CFU mL^−1^. Cell suspension (0.5 mL) of *Clostridium* was then mixed with 15 mL of semi-solid agar (0.4% agar, *w*/*v*) and poured to set a base agar. Agar discs containing different chemokines (10 mM saccharides, organic acids, amino acids; 2 g·L^−1^ soluble starch, peptone, yeast extract, ammonium salts, sodium salts, potassium salts; 0.8 g·L^−1^, 2.6 g·L^−1^ or 8 g·L^−1^ alcohols, phenol; 5 mM, 10 mM or 20 mM L-cysteine; 300 mg·L^−1^ or 1 g·L^−1^ glutathione) were placed on the base agar. These plates were incubated anaerobically at 37 °C for 12 h. Chemotaxis of *Clostridium* to a chemokine was indicated by the formation of cell rings around the agar discs [22].

### 2.6. Examination of Chemotaxis and Migration of Clostridium in Pit Mud

The culture of equally mixed 15 species of *Clostridium* (1 × 10^8^ CFU·g^−1^) was added to the sterile 10-year or 30-year pit mud and mixed thoroughly. These two pit mud portions were either stacked in parallel in a petri dish (method 1) or stacked in different plates (method 2). After 3 days of anaerobical cultivation at 37 °C, cell numbers of *Clostridium* in each portion or plate were determined to examine the migration of *Clostridium* within the pit mud. 

To detect the migration of *Clostridium* triggered by lactic acid, pit mud containing mixed *Clostridium* strains (1 × 10^8^ CFU·g^−1^) was divided into two portions, and they were stacked in parallel in a petri dish with addition of 2 g·kg^−1^ lactic acid (every 2 days) in one of them (method 3). Plates were incubated anaerobically at 37 °C for 11 days, cell numbers of *Clostridium* from both portions were determined at different time intervals. 

### 2.7. Migration and Metabolism of Clostridium in a Simulated Baijiu Fermentation System

Pit mud mixed with *Clostridium* was evenly smeared on the wall of 500 mL beakers. Sterile fermented grains (200 g) were then added in the center of beaker. Then 1 g·kg^−1^ of L-cysteine, acetic acid, glucose, sucrose or glutathione was added in fermented grains for every beaker, except for the blank. Finally, a layer of pit mud was covered on top of the fermented grains. Samples (fermented grains part) were collected and analyzed after 7 days of anaerobic fermentation at 37 °C. 

### 2.8. Analysis of Organic Acids

Pit mud and fermented grains samples were mixed with ddH_2_O at a ratio of 1:3 (*W:V*), centrifuged at 12,000 r·min^−1^ for 15 min, and filtered through a 0.22 μm water filtration membrane. The contents of lactic acid, acetic acid and butyric acid were determined by HPLC using the Aminex HPX-87H (7.8 mm × 300 mm) column. Column temperature: 40 °C; elution rate: 0.5 mL·min^−1^; UV detector: detection wavelength: 210 nm; injection volume: 10 μL; detection time: 30 min [21].

### 2.9. Construction of Phylogenetic Tree

The 16S rRNA sequences of isolated strains were submitted to NCBI database for comparison. Standard strains with close genetic relationship were selected according to the ID value. Phylogenetic tree was drawn by the Neighbor-joining method using MEGA X and iTOL v6.

### 2.10. Statistical Analysis

The experiments were repeated three times. SPSS 26.0 (version-26.0, IBM Corporation, Armonk, NY, USA)was used for one-way analysis of variance and significant difference analysis (*p* < 0.05). The histogram was prepared using Origin 2022b. The drawing pattern graph was carried out by Adobe Illustrator 2020 (version-2020, Adobe Systems Incorporated, San Jose, CA, USA).

## 3. Results

### 3.1. Isolation of Clostridium Strains from Pit Mud and Determination of Their Motility

A total of 172 bacterial strains belonging to clostridia, bacilli and actinobacteria were isolated from the pit mud. Among them, 110 strains were identified as *Clostridium* from 15 species (Figure 1, Table A1). Species including *C. tyrobutyricum*, *C. beijerinckii* and *C. butyricum* are commonly found in pit mud for the production of strong-flavor Baijiu from various regions in China [8,11]. Interestingly, three *Clostridium* species *C. tertium*, *C. pabulibutyricum* and *C. intestinale* were isolated from the strong-flavor Baijiu pit mud for the first time. In addition, 24 unclassified *Clostridium* strains (*Clostridium* sp.) were also isolated in this work. 

Strong-flavor Baijiu is produced through a solid-state fermentation. *Clostridium* is an obligate anaerobe and it is not the dominant genus in fermented grains. Thus, migration of *Clostridium* within the cell through motility is important for them to participate fermentation. The motility of *Clostridium* strains isolated from the pit mud were then examined and visualized on a semi-solid agar and confirmed by microscopic observation. It was found that most of the *Clostridium* strains (81%) isolated from strong-flavor Baijiu were motile (Table A2). Additionally, their motility that was exhibited on the agar resulted in various migration panels (Figure 2). This may be related to their differences in motile engines (flagella or pili) and chemotaxis.

### 3.2. Chemotaxis of Clostridium

In order to verify whether *Clostridium* can be attracted by chemokines that presented in fermented grains or pit mud, the chemotaxis of *Clostridium* strains to carbon and nitrogen sources, and volatiles including organic acids, alcohols and easters was determined. As shown in Figure 3a, chemotaxis of most *Clostridium* species to various saccharides including sucrose, maltose, fructose and glucose was observed. *Clostridium* sp. and *C. subterminale* have no chemotaxis to all the tested saccharides, and *C. pabulibutyricum* is chemotactic to fructose only. *Clostridium* sp., *C. pabulibutyricum* and *C. pabulibutyricum* only have chemotaxis to lactic acid, while other *Clostridium* species have chemotaxis to lactic acid, acetic acid and butyric acid. Peptone and yeast extract are not chemokines for the tested *Clostridium*. Most *Clostridium* strains have no chemotaxis to ammonium salt, except for *C. amylolyticum, C. sartagoforme* and *C. celerecresens* has chemotaxis to ammonium acetate, ammonium sulfate and ammonium phosphate, respectively. Most *Clostridium* species have chemotaxis to L-cysteine, glutamate and glutathione, and leucine is the chemokine to *C. intestinale, C. kogasensis, C. pabulibutyricum* and *C. beijerinckii*. Few *Clostridium* species have chemotaxis to valine and L-phenylalanine. *Clostridium* sp. and *C. subterminale* showed no chemotaxis to the tested amino acids. Interestingly, the diameter of chemotaxis rings increased with the increasing concentration (from 5 mM to 10 mM) of L-cysteine (Figure 3b). No *Clostridium* strains exhibited chemotaxis to the tested potassium salts and there were only a few *Clostridium* species that had chemotaxis to sodium salts. *Clostridium* exhibited no chemotaxis to alcohols when the corresponding compounds were added at the concentration of 0.8 g·L^−1^ and 2.6 g·L^−1^. However, they showed chemotaxis to 8 g·L^−1^ of isooctanol or 2-octanol (Figure 3b). This effect was not observed for other tested alcohols. 

Lactic acid bacteria (LAB), yeast and *Bacillus* are critical and functional microorganisms for strong-flavor Baijiu production. Growth and metabolism of *Clostridium* during strong-flavor Baijiu fermentation can be affected by these microorganisms. In this work, chemotaxis of *Clostridium* to LAB, yeast and *Bacillus* strains (isolated from pit mud or fermented grains of strong-flavor Baijiu) were investigated. It showed that none of the *Clostridium* had chemotaxis to the tested *Bacillus* or yeast strains (Figure 4a). However, all of them had chemotaxis to the tested LAB (Figure 4a,b). 

### 3.3. Migration of Clostridium within the Pit Mud

Previous work reported a significant difference in relative abundance of *Clostridium* in 10-year pit mud and 30-year pit mud, which was different in physiochemical properties [15]. These differences could be the factors (chemotaxis) that trigger the migration of *Clostridium* in pit mud. Thus, whether *Clostridium* could migrate within pit mud was investigated in order to characterize the chemotactic behavior of *Clostridium* in pit mud and understand their function in the brewing process. As shown in Figure 5a, the cell numbers of *Clostridium* in the 30-year pit mud portions were significantly higher than that in the 10-year ones for both culture methods. Considering the slight growth of *Clostridium* (method 2) in the pit mud, more cells of *Clostridium* were determined in the 30-year pit mud portion and less of them were detected in 10-year pit mud for culture method 1 than that for culture method 2. This indicated that a portion of *Clostridium* migrated from the 10-year pit mud to the 30-year pit mud through chemotactic movement under the condition of culture method 1. 

The motile behaviors of *Clostridium* in pit mud with or without a chemokine (lactic acid) were then investigated to further confirm the migration of *Clostridium* within pit mud through chemotaxis (culture method 3, Figure 5b). It was found that cells of *Clostridium* in the pit mud containing lactic acid were more than that in the pit mud without lactic acid. With the increasing cell numbers of *Clostridium* in the pit mud containing chemokine, cell numbers of *Clostridium* in the pit mud without chemokine decreased accordingly. Interestingly, contents of acetic acid and butyric acid in the pit mud after continuous supplementation of lactic acid were significantly increased, due to the enrichment of *Clostridium* and its synthesis of these organic acids by utilization of lactic acid (Figure 5c). 

### 3.4. Reduction of Lactic Acid Content in Fermented Grains by Clostridium through Chemotaxis

It was demonstrated that *Clostridium* could migrate within the pit mud triggered by chemotaxis and utilize nutrients (lactic acid) present in a different location, in the above section. In a strong-flavor Baijiu fermentation cell, most of the nutrients including chemokines are present in fermented grains. Additionally, one of the ways for *Clostridium* to participate in Baijiu fermentation is migrating to fermented grains and metabolizing compounds there. Thus, a simulated strong-flavor Baijiu fermentation system (Figure 6a) was employed to investigate the metabolism of *Clostridium* during fermentation and its migration crossing pit mud and fermented grains through chemotactic movement. As shown in Figure 6b, lactic acid is present at a high level (14.9 g·L^−1^) in the original fermented grains (the blank, Figure 6b). With the addition of *Clostridium* in pit mud, content of lactic acid in fermented grains significantly decreased to 12.3 g·L^−1^. Moreover, the addition of extra chemokines (L-cysteine, acetic acid, glucose, sucrose, glutathione) other than lactic acid in fermented grains resulted in the enhanced reduction of lactic acid. Lactic acid content in fermented grains was reduced by a maximum by 38.9%. This indicated that *Clostridium* could migrate from pit mud to fermented grains through chemotaxis and utilize nutrients in fermented grains. 

## 4. Discussion

*Clostridium* is a functional microorganism in pit mud, it can adapt to the brewing environment and participate in strong-flavor Baijiu fermentation. Pit mud from different Baijiu fermentation cells (different distillers or different pit ages) has unique physiochemical properties and is different in the microbial community. In this work, 110 *Clostridium* strains belonging to 15 species were isolated from the pit mud for strong-flavor Baijiu production. Their chemotaxis to various compounds such as saccharides, amino acids and organic acids were reported. The migrations of *Clostridium* within the pit mud and crossing pit mud and fermented grains triggered by chemotaxis were demonstrated for the first time. Isolation of novel *Clostridium* species from pit mud provides resources for working on pit mud microorganisms. Additionally, characterization of motile behaviors of *Clostridium* helps to understand their function in the process of Baijiu fermentation.

There are about 39 species of *Clostridium* that have been isolated from strong-flavor Baijiu pit mud [3,23,24]. However, the analysis of the microbial community of pit mud revealed that there were still many *clostridia* in pit mud had not been isolated, due to the heterogeneity and complexity of pit mud and the limited cultivation methods for *Clostridium*. In this work, *C. tertium*, *C. pabulibutyricum* and *C. intestinale* were isolated from the pit mud of strong-flavor Baijiu for the first time (Figure 1). We also obtained a number of *Clostridium* sp. strains. They might be potential *Clostridium* novel species. It was found that 81% of these strains were motile and they exhibited different motile panels (Figure 2). This may be related to their differences in motile engines or chemotaxis (Figure 3). 

Microbial community succession was commonly observed in the process of Baijiu fermentation [25]. Although *Clostridium* mainly exists in pit mud, it was detected in fermented grains during Baijiu fermentation [26,27,28]. The mechanism for the changes in *Clostridium* abundance in fermented grains is not clear. Chemotaxis of *Clostridium* could be one of the explanations as we observed this behavior for the pit mud isolates. As shown in Figure 3, the chemotaxis of 15 *Clostridium* strains to substances (sugars, free amino acids, organic acids, sodium salts) that commonly present in pit mud or fermented grains were detected. The chemotaxis of *Clostridium* to these compounds may be the reason of enhancing the enrichment of *Clostridium* in an anaerobic fermentation with lactic acid and acetic acid as carbon sources [16]. Previous work reported that phosphorus and potassium were positively correlated with *Clostridium* [15,29]. Our work did not find the connection of this with chemotaxis of *Clostridium*. Interestingly, all *Clostridium* strains exhibited chemotaxis to tested *Lactobacillus* strains (Figure 4a). This may be related to the production of lactic acid and acetic acid by LAB, as these organic acids are proven to be chemokines for *Clostridium* (Figure 3a). Although yeast may provide anaerobic environment for *Clostridium* to promote their growth and metabolism by consumption of oxygen [3,30,31], no chemotaxis of *Clostridium* to tested yeast strains was observed (Figure 4a). 

Chemotaxis of bacteria is mainly mediated by the movement of bacteria. This is greatly affected by both physical (i.e., water content, medium pore size, pH value, temperature) and chemical factors (nutrients, oxygen, etc.) [5,32,33]. The migration of *Clostridium* from 10-year pit mud to 30-year pit mud indicated that mature pit mud was more suitable for *Clostridium* than growing pit mud (Figure 5a). In addition, we demonstrated that single chemokine could trigger the migration of *Clostridium* within pit mud or crossing pit mud and fermented grains (Figure 5b and Figure 6). In a stimulated strong-flavor Baijiu fermentation system, chemokines such as sucrose and acetic acid significantly promoted the migration of *Clostridium* from pit mud to fermented grains and the utilization of lactic acid in fermented grains (Figure 6b). This work demonstrated the successful enrichment of *Clostridium* and promotion of its corresponding metabolism (utilization of lactic acid) in fermented grains. It provides practical examples for control and regulation of metabolism and migration of *Clostridium* in the process of Baijiu fermentation.

## Figures and Tables

**Figure 1 foods-11-03639-f001:**
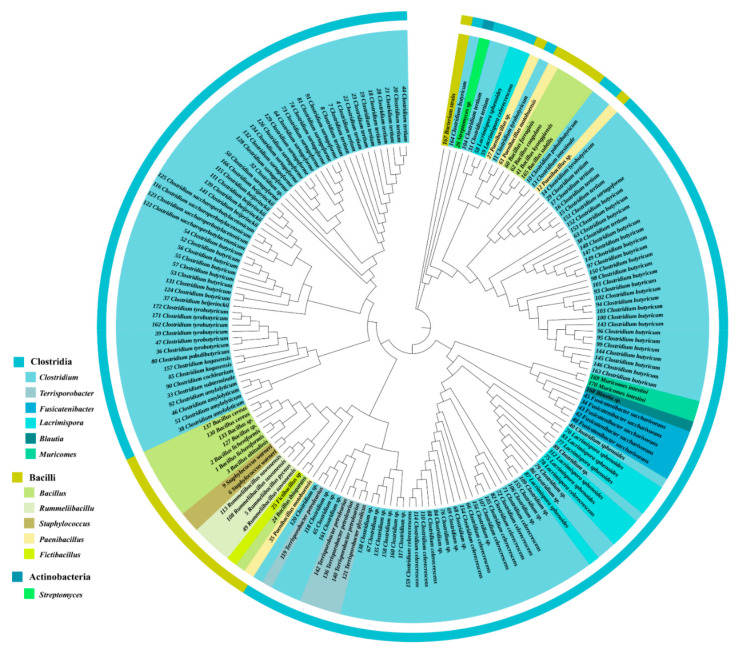
Phylogenetic tree of pit mud isolates constructed by the Neighbor-joining method using the 16S rRNA gene sequences.

**Figure 2 foods-11-03639-f002:**
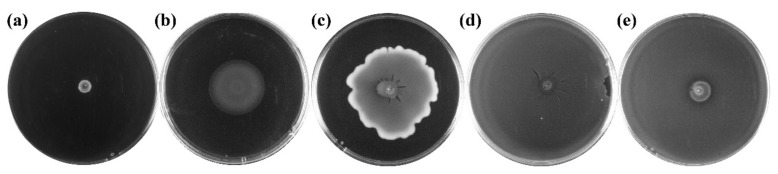
Motility panel of *Clostridium* strains isolated from pit mud: (**a**) non-motile; (**b**–**e**) motile.

**Figure 3 foods-11-03639-f003:**
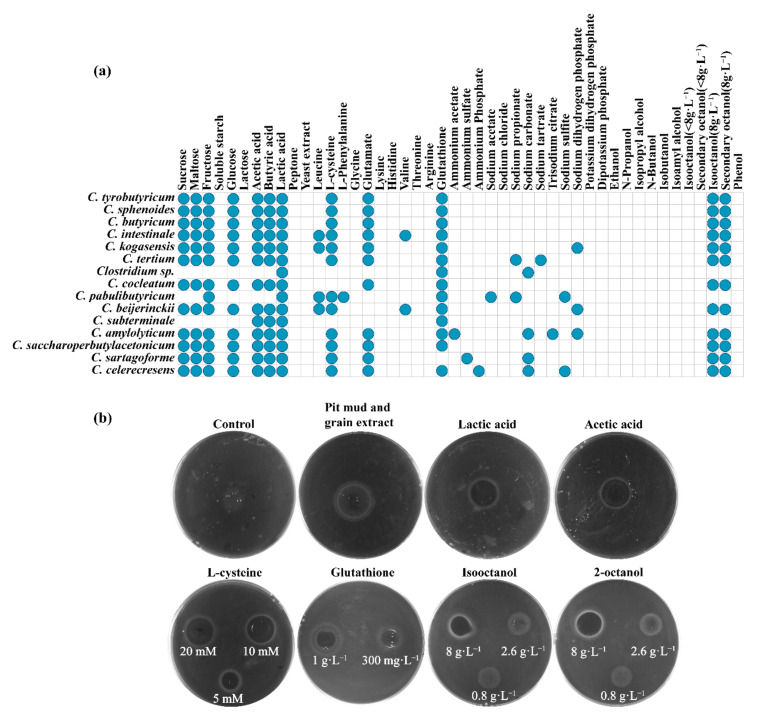
Analysis of chemotaxis of *Clostridium* strains to chemical factors: (**a**) chemotaxis of *Clostridium* strains to different substances; (**b**) chemotaxis of *Clostridium* detected by the agar disc method.

**Figure 4 foods-11-03639-f004:**
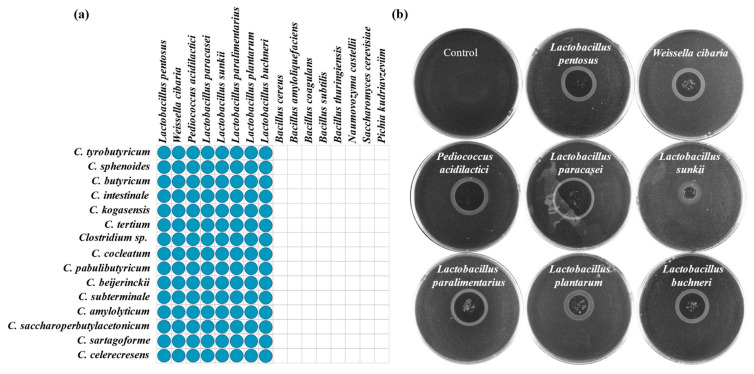
Chemotaxis of *Clostridium* to microorganisms that involved in Baijiu fermentation: (**a**) chemotaxis of *Clostridium* to LAB, Bacillus and yeast; (**b**) detection of chemotaxis of *Clostridium* to LAB using the agar disc method.

**Figure 5 foods-11-03639-f005:**
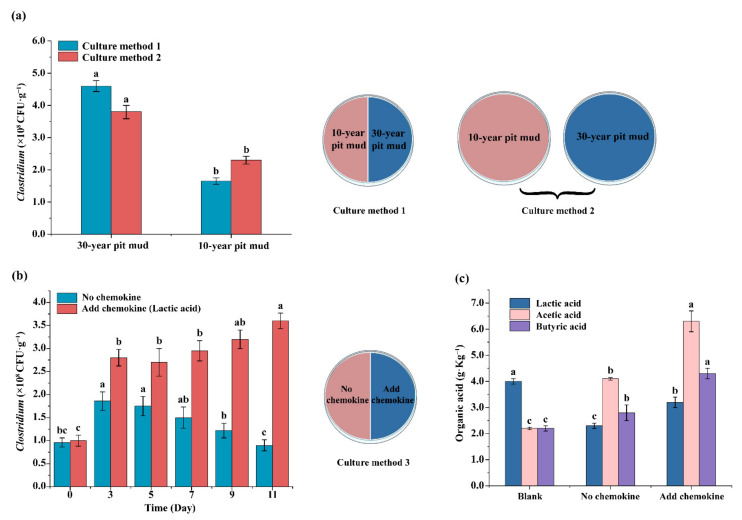
Migration of *Clostridium* in pit mud triggered by chemotaxis: (**a**) migration of *Clostridium* from growing pit mud to mature pit mud; (**b**) migration of *Clostridium* within pit mud triggered by lactic acid; (**c**) comparison of organic acids content in different pit mud samples. Letters “a–c” indicate the significant differences (*p* < 0.05).

**Figure 6 foods-11-03639-f006:**
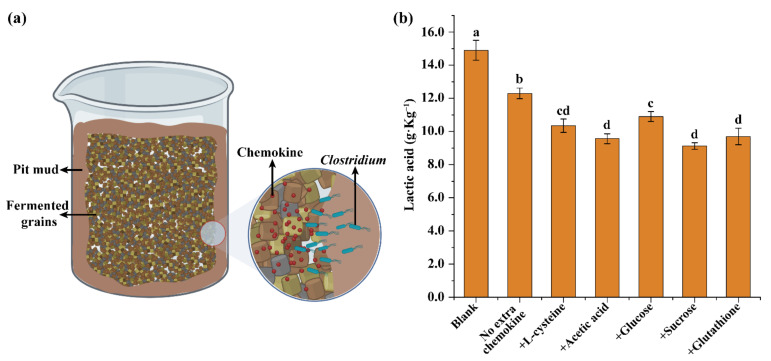
Reduction of lactic acid content in fermented grains by *Clostridium* from pit mud: (**a**) scheme of simulated strong-flavor Baijiu fermentation; (**b**) reduction of lactic acid in fermented grains by *Clostridium* through chemotaxis. Letters “a–d” indicate the significant differences (*p* < 0.05).

## Data Availability

All data included in this study will be available by contacting the corresponding author as per the guidelines of the journal under the agreement.

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
