# Peer review of "Chemotaxis of *Clostridium* Strains Isolated from Pit Mud and Its Application in Baijiu Fermentation"

_foods, 2022, doi:10.3390/foods11223639_

Round 1
Reviewer 1 Report
This manuscript is about Clostridium strains belonging to 15 species were isolated from the pit mud for strong-flavor Baijiu production. Their chemotaxis to various compounds such as saccharides, amino acids and organic acids were also reported.
This paper is original, and interesting for the readers. It is written in good English, with very well-prepared figures to enhance the results. It is easy to be accessble for general readers. The conclusions of this paper are consistent with the evidence presented, and address the main question the author posed.
It is a high-quality manuscript, but some minor errors need to be fixed. i.e. 3.2 Chemotaix should be corrected to chemotaxis.
Author Response
The typos have been corrected as suggested.
Reviewer 2 Report
Authors have studied the chemotaxis of Clostridium from pit mud of Baijiu fermentation, including the identification of microorganisms. The strategies, methodologies, and results are written very well, logically, and scientifically.
A minor aspect of this manuscript is as follows;
In Figure 1, the names of microorganisms in the large circle might be not able to be written. Please think again and revise the presentation of Figure 1 as shown in this manuscript.
Author Response
Figure 1 is redrew as suggested.
Reviewer 3 Report
Dear Author:
The strong point of the manuscript entitled “Chemotaxis of Clostridium strains isolated from pit mud and its application in Baijiu fermentation” is the novelty of the used microbiology and the subtract. I think that figures, graphs and photos are of good quality and the methodology is appropriate.
Best Regards.
Author Response
Thanks for the reviewer's comments.